# Cell Death by Entosis: Triggers, Molecular Mechanisms and Clinical Significance

**DOI:** 10.3390/ijms23094985

**Published:** 2022-04-30

**Authors:** Mostafa Kianfar, Anna Balcerak, Mateusz Chmielarczyk, Leszek Tarnowski, Ewa A. Grzybowska

**Affiliations:** Maria Sklodowska-Curie National Research Institute of Oncology, Roentgena 5, 02-781 Warszawa, Poland; mostafa.kianfar@pib-nio.pl (M.K.); anna.balcerak@pib-nio.pl (A.B.); mateusz.chmielarczyk@pib-nio.pl (M.C.); leszek.tarnowski@pib-nio.pl (L.T.)

**Keywords:** programed cell death, entosis, cell-in-cell, actomyosin contractility, autophagy

## Abstract

Entosis—a homotypic insertion of one cell into another, resulting in a death of the invading cell—has been described in many reports, but crucial aspects of its molecular mechanisms and clinical significance still remain controversial. While actomyosin contractility of the invading cell is very well established as a driving force in the initial phase, and autophagy induced in the outer cell is determined as the main mechanism of degradation of the inner cell, many details remain unresolved. The multitude of triggering factors and crisscrossing molecular pathways described in entosis regulation make interpretations difficult. The question of the physiological role of entosis also remains unanswered. In this review, we summarize the knowledge of molecular mechanisms and clinical data concerning entosis accumulated so far, highlighting both coherent explanations and controversies.

## 1. Introduction

Entosis is a form of cell death that occurs when one cell inserts itself into the neighboring cell, which results in the ultimate death of the invading cell. This creates a characteristic cell-in-cell (CIC) pattern, observed as early as 1891 by Steinhaus [1] in tumor samples. CIC structures may result from several similar phenomena (see Table 1), but in this review, we will focus on entosis, which represents homotypic invasion—occurring between cells of the same type. Entosis represents suicidal but non-apoptotic cell death, governed by different principles and mechanisms than apoptosis or other types of programed cell death (see Table 2). Interestingly, an entrapped inner cell stays viable for some time and is even able to divide inside the host cell or escape outside, provoking questions about the regulation of the particular stages of the process. To date, entosis remains a relatively less known form of cell death, and many issues are still controversial, including the main question of whether it is pro- or anti-tumorigenic—or both, depending on the circumstances. In this paper, we describe the known triggers, mechanisms and stages of entosis and discuss its possible significance in development and cancer.

## 2. Mechanical and Structural Aspects of Entosis

### 2.1. Rho Signaling

Mechanically, entosis depends on actomyosin contractility regulated by the Rho-GTPase activity in the invading cell. The Rho family of small GTPases comprises many proteins, but the most important three members include Rho (RhoA, RhoB and RhoC), Rac (Rac1, Rac2 and Rac3) and Cdc42. These three proteins orchestrate cell adhesion and migration, acting as molecular switches with active GTP-bound and inactive GDP-bound conformations. Activated Rho-GTP activates its major downstream effector, Rho-associated protein kinase ROCK, which leads to phosphorylation of myosin light chain (MLC) and suppression of MLC phosphatase, promoting actin–myosin II interaction and an increase in contractility, actin stress fiber formation and focal adhesion. Cell–cell junctions recruit Rho-GAPs (RhoGTPase activating proteins, which promote GDP-bound inactive conformation of Rho), locally inhibiting Rho activity and creating a gradient of active Rho in the cell. The polarized distribution of active RhoA in entotic internal cell at the rear end opposite to cell–cell junctional interface creates mechanical tensions, leading to internal cell invasion [2]. Accordingly, Purvanov et al. [3] observed that the invading cell displays plasma membrane blebbing followed by actin assembly at the rear, forming a uropod-like, actin-rich structure, which promotes invasion.

### 2.2. Cytoskeleton Involvement

Cell–cell contacts, cell-matrix adhesion and the connecting cytoskeleton act in concert, since matrix adhesion counterbalances cell–cell adhesion and inhibits entosis. Cytoskeleton dynamics plays a huge role in entosis as a scaffold, linking adhesion sites, but also as an active signaling hub. Wang et al. [4] described a multi-molecular complex termed a mechanical ring (MR) positioned between the invading and the engulfing cells. MR links adherens junctions with contractile actomyosin, coordinating their actions and promoting entosis. MR is highly enriched in vinculin, which serves as a mechanical sensor.

Actin filaments and microtubules act in opposite directions, resulting in cell contraction or expansion, respectively [5]. The entry of the inner cell is obviously dependent on actin filaments and actomyosin contractility, which is also essential for differential stiffness of the inner and outer cell and represents the driving force of the first phase of entosis (invasion). Actin cortex organization determines the cortical tension, resulting in local contractions and cellular deformations [6]. However, the microtubule cytoskeleton is also implicated in the regulation, as described by Xia et al. [7]. The authors have shown that microtubule plus-end dynamic is regulated by the phosphorylation of mitotic centromere-associated kinesin (MCAK) by Aurora-A kinase. The microtubule (MT) plus-end tracking protein TIP150 interacts with MCAK, regulating the dynamics of the kinetochore microtubule plus-end in-cell division [8]. It appears that the interaction of TIP150 with MCAK is modulated by Aurora-A phosphorylation of MCAK and affects entosis, possibly by changing cell rigidity, which is a factor differentiating invading cells and host cells. Apparently, cell rigidity—a net result of cytoskeleton dynamics—plays an important role in entosis, with stiffer cells (with higher elastic modulus, or Young’s modulus) invading softer cells (with lower elastic modulus) [9,10]. Interestingly, a process akin to entosis was described in mice embryo, and the differential tension has been demonstrated to suffice to recapitulate the process [11].

## 3. Triggers: Is There a Coherent Picture?

The described triggers pertain to the first phase of entosis, i.e., the entry of the inner cell into the host cell. To date, several triggering factors have been described, sometimes so different that the reporting authors had problems classifying the type of cell death they have encountered. In the following section, the main triggers are described along with the underlying pathways that were uncovered.

### 3.1. Matrix Detachment

In the seminal paper in Ref [2], Overholtzer et al. describe the process of entosis as a form of cell death triggered by matrix detachment of the epithelial breast cancer cells. They provide evidence that this is non-apoptotic (proceeding without nuclear fragmentation or caspase activation), hence also non-anoikis cell death, which is triggered by matrix detachment and occurs in human tumors. They coined the term “entosis” (from Greek “entos”, which means “within”) and provided the description of the process as requiring actin, myosin II, Rho and ROCK activity and occurring with lysosomal degradation of the internalized cell. They have also shown that the process can be, to some point, reversed, and the internalized cell can be released. They also primarily characterized the role of cell–cell junctions, and especially cadherins, in entosis. Importantly, this work was carried out on cells in suspension, despite the fact that the breast cancer cell line (MCF7) is adherent because entosis achieves much higher levels under these conditions. Interestingly, subsequent studies by Wan et al. [12] have shown that matrix detachment is, in fact, not necessary for entosis, but the authors of this second report confirm the role of the Rho-ROCK-actomyosin pathway in entosis induction. They explored the effects of the interplay between the two polarity proteins, partitioning defective 3 homolog (Par3) and mammalian homolog of Drosophila Lethal (2) Giant Larvae (Lgl1/2), on myosin II activation, coming to the conclusion that the resulting unbalanced myosin II activity is a driving force for entosis.

### 3.2. E-Cadherin Expression

E-cadherin expression was demonstrated to induce entosis in MCF7 cells in suspension by Sun et al. [13]. Again, the mechanism was shown to be associated with the polarized RhoA activity (regulated by p190A RhoGAP activity), phosphorylation of myosin light chain 2 and the activation of contractile myosin in the majority of internalizing cells compared to the hosts. This is consistent with the finding that IL-8 depletion caused significant reduction in the expression of CDH3 (P-cadherin) and CTNNG (junctional plaque component, plakoglobin), resulting in inhibition of entosis, while IL-8 treatment reversed this phenotype [14].

### 3.3. Stress Factors

#### 3.3.1. Metabolic Triggers: Glucose and Amino Acid Starvation

Nutrient starvation has been shown to play a pivotal role in cell engulfment and inducing entosis [10]. While it has been shown that amino acids recovered from entotic inner cells contribute to nutrient balance [15], subsequent studies revealed that amino acid deprivation did not promote entosis as efficiently as glucose starvation [10]. Moreover, recently, it has been demonstrated that amino acid starvation led to decreased rates of entotic cell death, independent of apoptosis, acting through the mTOR pathway [16]. Interestingly, the regulation by mTOR is independent of autophagy activation, despite the fact that autophagy was shown to play a role in the elimination of the entotic inner cell. Typically, inhibition of mTOR is the key factor in autophagy induction. Instead, the described mTOR-mediated regulation is autophagy independent and involves the 4E-BP1/2 proteins, which are known regulators of mRNA translation [16].

Glucose starvation was shown to enhance entosis by activating AMP-activated protein kinase (AMPK) in the internal cells, as described by Hamann et al. [10]. AMPK, the ATP/AMP sensor, regulates cellular energy homeostasis by activating glucose uptake and other metabolic pathways when the cellular energy balance is low. The authors have shown that AMPK activation enhances the frequency of entosis even in the presence of glucose. Many questions remain unanswered: Why is AMPK activated in some cells and not others in a homotypic cell population? Are cells that activate AMPK more energetically stressed? How is it linked to the mechanotransduction of tensions, which are crucial for entosis? The authors in Ref [10] partially address this last question, providing evidence that glucose starvation results in changes in cell deformability, leading to bimodal distribution of cells; upon starvation, two populations of cells emerge with low and high elastic modulus. This situation directly leads to entosis; however, the exact mechanisms behind it remain unclear. Some insight may be provided by the observation that AMPK activation can be induced by the application of shear stress or force on E-cadherin at cell–cell junctions. Upon mechanical stress, AMPK is recruited to adhesion complexes, and its activation leads to increased phosphorylation and activation of myosin II [17,18,19]. This linking of mechanotransduction and metabolism may be a key factor in understanding entosis.

Interestingly, Hamann et al. [10] also observed that in glucose starvation-induced entosis, entotic cells died more rapidly and escaped less frequently than with some other triggers, suggesting differences in the regulation of some particular steps of the post-invasion phase.

#### 3.3.2. Ultraviolet Radiation

In a recent report, Chen et al. [20] describe one more factor; ultraviolet radiation was found to activate JNK and p38 stress-activated kinase signaling and, in turn, induce entosis. In UV stressed cells, entosis activation appears to be a part of the mixed-cell death response with parallel induction of apoptosis and necrosis, and some interplay between these different forms of cell death was implied (also observed with TRAIL-induced entosis, described below).

### 3.4. Mitosis

Mitotic entosis was first described by Durgan et al. [21] as regulated by small GTPase cell division control protein 42 (Cdc42) involved in the signaling pathways linked to actomyosin organization and shaping cell morphology, cell migration and cell cycle progression [22]. The authors have shown that Cdc42 depletion augments mitotic cell rounding and associated deadhesion, triggering entosis, and that these effects depend on RhoA activation. Cdc42 and RhoA both represent small GTPases from the Rho family involved in the regulation of cytoskeletal dynamic but with opposing spatio-temporal signaling programs. Moreover, it has been demonstrated that the cell stiffens during mitosis [23], which tallies with the mechanistic observation that the more rigid cell invades the more elastic cell. Most interestingly, this type of entosis occurred in adherent but not suspended cell cultures, suggesting the existence of the distinct mechanisms that trigger the process in both conditions. Despite this, adherent entosis observed upon Cdc42 depletion shares the mechanistic features of entosis in suspension, namely, the involvement of RhoA/ROCK pathway, myosin II activation and non-canonical autophagy with lysosomal degradation of the internalized cell. The authors observed a clear relationship between cell division and entosis in Cdc42-depleted cells using a cell-cycle inhibitor (Cdk1 inhibitor), which arrests cells at the G2/M boundary, and obtaining profound decrease in entosis. Interestingly, cell-cycle arrest performed with taxanes caused the opposite effect, that is, an increase in entosis in the treated cells. The reason for this discrepancy lies in a different mode of action of both inhibitors; Cdk1 inhibitor arrests cells in G2/M, inhibiting entry into mitosis, while taxanes arrest cells in prometaphase, which increases the number of round mitotic cells.

### 3.5. Ligand-Mediated Entosis

#### 3.5.1. Lysophosphatidic Acid (LPA)

Purvanov et al. [3] demonstrated that LPA—a potent mitogen—promotes entosis in MCF10A cells via G-protein-coupled receptor LPAR2. LPAR2 signals through heterotrimeric G proteins, inducing PDZ-RhoGEF and RhoA activation, which is in line with previously described mechanisms. The authors also observed that Diaphanous formin mDia1 is necessary for entosis downstream of LPAR2 and that it functions by spatially controlling polarized actin assembly, which is crucial for entosis.

#### 3.5.2. TRAIL Signaling

TNF-related apoptosis-inducing ligand (TRAIL) is a member of a large TNF superfamily of factors that can trigger apoptosis. Bozkurt et al. [24] have shown that TRAIL activates not only apoptosis but also entosis in colon cancer cells. The initiation of both of these cell death programs requires TRAIL receptors DR4 and DR5, but the signaling diverges at caspase-8. Caspase-8 appears to be required for TRAIL-induced entosis but not as an enzyme, since the catalytically inactive CASP8 mutant restores entosis in a CASP8 −/− background. Interestingly, the authors observed that inhibition of apoptosis increases the rate of inner cell release, which offers some insight into the mechanisms behind the decisions concerning the fate of the inner cell.

## 4. Discussion Points

### 4.1. Are Those Triggers at Some Point Related?

While the triggers seem to be diverse, and the list is probably still incomplete, one may wonder if there is a common denominator downstream of these factors. For now, it seems that the most universal element is represented by a mechanistic explanation, involving Rho/ROCK signaling and myosin II activation as the endpoint of this induction phase. While in some cases the link is not obvious (for example, with glucose starvation), the accumulated results suggest that various possible triggers ultimately activate myosin II and start entosis. There is a universal question, which might be asked regarding all these triggering factors: why, in a homogenous cell population, do the triggers/stress factors elicit different effects, prompting an invasion of one neighboring cell into another? Does it depend on the cell-cycle stage or the level of stress or individual resistance? Does it explore minimal heterogeneity, present in every, even clonal, cell population? Future research should address and resolve these doubts.

### 4.2. Entosis in Cells in Suspension vs. Monolayer

Since entosis was first described as a process affecting individual cells in suspension [2], many subsequent studies were performed in this model. The use of this model is also justified by the fact that entosis in these conditions is much more intense than for the adherent cells, achieving around 30% or more. However, one may ask if this model is physiologically correct, since the MCF7 breast cancer cell line used in these experiments is strongly epithelial and adherent. Moreover, if we would like to relate the cell line model to the situation in tumors, one may argue that adherent cell layer is more relevant. Entosis in adherent cells was reported relatively recently [10,21], although some earlier reports highlighted this possibility [25]. It is not surprising that it was spotted later, since it is minimal—around 1–1.5% of the cell population. Apparently, as pointed out in a chapter concerning mitosis-induced entosis, the mechanisms of entosis for cells in suspension and for adherent cells may differ—for example, entosis triggered by Cdc42 depletion could be observed in adherent culture but not in cells in suspension. In line with these observations, while Xia et al. [7] reported a substantial decrease in entosis after taxol treatment for cells in suspension, Durgan et al. [21] observed increased entosis after similar treatment in adherent conditions. The induction of entosis by E-cadherin expression described by Sun et al. [13] can also be specific to matrix-detached conditions, since adherent epithelial cells already have strong cell–cell contacts, therefore, an increased expression of E-cadherin may not be as effective.

Furthermore, Garanina et al. [26] suggested that the outer cell has a more active role in entosis in adherent culture than in cells cultured in suspension.

Overall, it is not surprising that the mechanisms of entosis may differ in these two conditions, since, from a mechanistic point of view, cells are subjected to very different tensions and forces, and their shape is different when they are free floating and when they are adherent, when they form small patches or grow as a tightly packed monolayer. It would be interesting to establish the differences and similarities in entotic mechanisms under detached/adhering conditions. For now, myosin II activation seems to be a common factor common to all these models.

### 4.3. Cell Competition Interpretation

Sun et al. [9] proposed that entosis represents a competition between the engulfing “winners” (since they stay alive and benefit from the nutrients) and the invading “losers” (since they ultimately die). This optic, however, is a bit askew because the “winner” role is passive, so, although they outcompete the “losers” and dominate the cell population, this is rather a net result of a complicated phenomenon. The authors tested heterotypic entosis between cells from different cell lines, noticing a correlation between the “winner” status and the increased mechanical deformability. Interestingly, the authors also associate a “winner” status with Kras activation, suggesting that it may inhibit actomyosin contraction and promote deformability. This may be hard to reconcile with the fact that mitosis was reported as the inducing factor, and in Kras-transformed cells, the proliferation/mitotic index is higher—however, as stated before, these two experiments were performed in different conditions (detached/adherent), so the mechanisms may also differ.

### 4.4. Entosis vs. Phagocytosis

As pointed out by Florey et al. [27] in a very informative quick guide to entosis, dead cell clearance by phagocytosis, despite similarities in appearance, is completely different. First, phagocytosis is heterotypic, with immune cells (e.g., macrophages or dendritic cells) eliminating apoptotic epithelial (or other) cells. Second, in phagocytosis, the engulfing cell represents the active participant of the process, responding with cytoskeletal rearrangements to the “eat-me” signals on the apoptotic cell surface (mainly phosphatidylserine). In entosis, the cytoskeletal rearrangements take place in the invading internal cell. Thus, entosis rather resembles an active pathogen cell invasion, occurring in non-phagocytic cells [28]. One more difference is that while phagocytosis comprises dead cell elimination, in entosis, the eliminated cells are fully viable.

## 5. Initiation and Execution of Entosis

Entosis has two very distinct phases with a characteristic pause between them: the initiation phase, leading to the formation of cell-in-cell structures, and the execution phase, proceeding with the degradation of the inner cell. The first phase, in which the invasion occurs, is obviously dependent on actomyosin contractility. The second phase is related to autophagy. The general scheme of these phases is depicted in Figure 1.

### 5.1. Initiation

The conditions and mechanisms leading to initiation were described above, with the main driving force being represented by the activation of the Rho-ROCK pathway and actomyosin contractility in the invading cell, together with differences in deformability and rigidity of the host cell and the invading cell. As shown by several studies, these factors can be linked to metabolic or proliferative status of the cell, as well as the level of cellular stress. It is important to reiterate that the inner cell is actively engaged in the initiation of the whole process; however, in the adherent monolayer, the outer cell may play a more active role. Garanina et al. [26] observed that microtubules and the Golgi apparatus were involved in the increase in the outer cell surface area associated with the formation of the membrane protrusion necessary to engulf the inner cell.

The hiatus between initiation and execution is very interesting from the perspective of the regulation and the outcome of the whole process, since it is known that the demise of the inner cell is not inevitable. The internalized cell can be viable for many hours and may escape outside. As reported by Overholtzer et al. [2], 12–18% of internalized cells were eventually released in MCF7 and MCF10A cell culture, respectively. These cells appear normal after release and are able to divide. Moreover, a small percentage of internalized cells can divide inside the host cell (9% and 0.8% for MCF7 and MCF10A, respectively). The most interesting questions are why there is this lengthy indecision period, and when and why does the cell reach the point of no return?

### 5.2. Execution

Garanina et al. [26] described five consecutive stages of entosis in adherent cell monolayer after cell-in-cell structure formation. This classification is based on an examination of the morphological changes during entosis, using scanning electron microscopy and phase contrast microscopy. The authors have observed that in the first stage, the inner cell is round, with a round and unchanged nucleus. The size of the inner cell is similar to the size before internalization, and its plasma membrane is close to the entotic vacuole, with cell–cell junctions present. In the second stage, the cell shrinks and the vacuole enlarges. In the third stage, the internalized cell acquires an irregular shape and displays chromatin condensation. In the fourth stage, the cell and its nucleus are deformed, and the nuclei disappear. In the fifth stage, all that remains of the inner cell is vacuolized cytoplasm and condensed chromatin.

As shown by Florey et al. [27], at the execution phase, the inner cell is eliminated by selective autophagy. Autophagy is a controlled self-destruction program that serves as a nourishing mechanism induced by starvation (non-selective autophagy) but also an important mechanism of elimination of damaged organelles and protein aggregates (selective autophagy, pertaining only to some specific cargo). In classic, starvation-induced autophagy, part of the cell is separated from the rest of the cytoplasm by a double-membrane structure called autophagosome, which is subsequently lysed to the lysosome and degraded. A variation of this program called xenophagy is engaged in the removal of bacterial cells that invaded the non-phagocytic mammalian cells. It has been demonstrated [29] that autophagy is also employed in the elimination of entotic internal cells.

Entotic cell death uses autophagic machinery but independent of the double-membrane autophagosome formation [29]. In an unconventional (but not unprecedented) manner, the single-membrane entotic vacuole is decorated by the Light Chain 3 (LC3) protein lipidated by phosphatidylethanolamine (PE). LC3-PE, also called LC3-II, represents a known autophagy marker. Autophagy machinery proteins Atg5, Atg7 and class III PI-3-kinase Vps34 are also engaged in this process, but Fip200, a ULK-interacting protein (Unc-51-like autophagy activating kinase) involved in autophagosome formation, has no effect on LC3 recruitment to the entotic vacuole, showing that this pathway differs from the standard, starvation-induced pathway and rather resembles pathogen elimination. Subsequently, LC3-decorated entotic vacuoles lyse with lysosomes, leading to the acidification and degradation of internalized cells. In effect, the entotic inner cell is eliminated using autophagy machinery, but this process is not autophagosome or mTOR dependent, so it represents quite an unusual type of autophagy.

Interestingly, while apoptotic machinery is not induced during entosis [2], when autophagy is inhibited or when lysosome is not operational, entotic cell death switches to apoptotic [2,29].

The important question that remains is why are some cells degraded and some escape? Is the decision dependent on the internal cell, external cell or the crosstalk between them? May it depend on the autophagy mechanisms?

## 6. What Is Entosis for? Developmental and Clinical Implications

The physiological role of entosis still requires explanation. This role may include the clearance of unwanted cells in processes such as embryo development or cancer progression, but other explanations have also been proposed.

### 6.1. Embryo Development

The role of apoptosis in the elimination of cell surplus during embryo formation and sculpting is well established; however, it turns out that the other mechanisms may also be employed. Lee et al. [30] have demonstrated that entosis eliminates the male-specific linker cell in C. elegans. Elimination of the linker cell is required for the joining of the male gonad and the cloaca, which forms the exit route for the sperm [31]. It was shown that linker cell death cannot be attributed to apoptosis and that it proceeds with the engulfment by the neighboring cell (U cell) and that the engulfed cell is viable for some time. It was demonstrated that linker cell clearance involves cell–cell adhesions and proceeds in an actin-dependent manner, further pointing to entosis [30]. Furthermore, morphological changes during linker cell death resemble those described for entosis: indentation (crenellation) of the host cell nucleus and vascularization of the linker cell [30,31]. The only problem with the classification of this specific cell death as entosis is that some early data indicated that autophagy is not involved in the clearance; vesicles were shown to be autophagosomes, the recruitment of LC3 (LGG1 in C.elegans) to the vesicular membrane was minimal (only 1.5-fold increase), and there was no effect on linker cell survival in some autophagy mutants [31]. However, further research confirmed the fusion with lysosomes [30] and, as pointed out earlier, entotic vacuole is actually different from autophagosome. In effect, while some classic autophagy signs may not be present, entotic autophagy is selective and specific, so the observed cell death can be classified as such.

This one example is obviously not enough to assume a significant role of entosis in development, but it might be the first of many.

### 6.2. Role in Cancer

The physiological role of entosis in cancer is puzzling, with many conflicting data and different interpretations. One of the proposed explanations assumes the role in the elimination of proliferating tumor cells; the finding that mitosis is the inducing factor for entosis particularly justifies such assumptions. However, entosis was found to induce aneuploidy and genomic instability in host cells due to a deformation of the host cell nucleus, which interferes with the subsequent cell division [32,33]. Mackay et al. [33] suggested that p53 status confers heterogeneity in a mixed cell population, leading to entosis. They have observed that wild-type p53 host cells died due to aberrant cell division, while more resilient mutant p53 host cells survived despite the aberrant divisions, multinucleation and tripolar mitoses. They have proposed that p53 mutant expression promotes tumorigenesis and genome instability via entosis. However, there are some issues with this explanation. In the proposed cell engulfment process, the host cell—not the internal cell—ends up dead, which is not typical for entosis. Secondly, Liang et al. [34] have also studied the role of p53 in entosis, using wild-type p53 and knockdown cells but not mutants; interestingly, they came to different conclusions regarding the role of entosis in cancer. In their studies, they proved that p53 assists the internalization and elimination of cells arrested in the metaphase. Such prolonged mitotic arrest may lead to aneuploidy, but this time it will be an aneuploidy of internal entotic cells, not the host cells. Damaged, aneuploid internal cell is cleared via p53-dependent entosis, which is mediated by Rnd3, a p53 target gene. Rnd3 is also a Rho GTPase that inhibits RhoA signaling by targeting both ROCK1 and p190A RhoGAP, which was proposed to compartmentalize RhoA activity, leading to activation of actomyosin in the rear. Therefore, they have characterized entosis as a surveillance mechanism to safeguard genome integrity and counteract tumor promotion.

There are some indications pointing to the role of chemotherapy in inducing entosis [35,36,37]. The interpretation of this phenomenon as pro- or anti-tumorigenic is not very clear at the moment, but most of the authors opt for its role in survival advantage for the engulfing cell.

While there are still a lot of questions concerning the role of entosis in cancer at the molecular level, the enumeration of entosis was studied in several cancers for its prognostic value. The published clinico-histopathological studies of entosis mostly associate its presence with a more malignant phenotype and worse prognosis (summarized in Ref [38]).

High numbers of entosis were shown to positively correlate with better survival for anal cancer [39] and some subtypes of breast cancer [40], while low numbers had a beneficial prognostic value in rectal cancer [39], head and neck squamous cell carcinoma (HNSCC) [39,41], lung adenocarcinoma [33] and pancreatic ductal adenocarcinoma (PDAC) [42]. In HNSCC, PDAC and some subtypes of breast cancer entosis was shown to be an independent prognostic factor.

Breast cancer is especially interesting, as entosis is mostly studied in breast cancer cell lines. Breast cancer is highly heterogeneous and has many biological and molecular subtypes, classified according to the expression of ER, PR and Her2 receptors and Ki-67 proliferation marker. Xin et al. [40] provided subtype-based analysis assessing the prognostic value of entosis in early breast cancer. Their results indicate that CIC/entosis profiling is a valuable tool for the prognostic analysis of breast cancer, but they also reveal that entosis is actually a favorable factor for the prognosis, especially in the luminal B Her2+ subtype. For other subtypes, the results were different but not statistically significant. This result is contrary to previous indications that entosis in tumor samples correlates with breast cancer malignancy [43]; however, this second study is much larger (148 vs. 50 samples) and subtype based, so it is more reliable.

In general, although entosis was established as a marker associated with a worse prognosis for some tumors, it seems to be tumor or even subtype dependent, so there is no clear recommendation on how to interpret it.

The enumeration of entosis in tumor samples can be used for prognostic purposes, but one has to keep in mind that even if there is a positive correlation of entotic numbers with higher histological grade or shorter overall survival, the interpretation is not straightforward; it may signify that entosis intensifies progression, but it may also imply a protective response to uncontrolled tumor cell proliferation.

## 7. Conclusions

Although cell death by entosis is very characteristic and has been described in many reports, there are still many gray areas pertaining to both the molecular mechanisms and the clinical significance of this process. Microscopic imaging is still the only way to identify and quantify entosis, and there are no specific markers characterized so far, so entosis cannot be easily measured in a simple, standardized assay, such as, for example, apoptosis. However, the knowledge accumulated so far implies that entosis may be of utmost importance as a method of controlled epithelial cell elimination, suggesting its possible significance in embryo development and cancer.

## Figures and Tables

**Figure 1 ijms-23-04985-f001:**
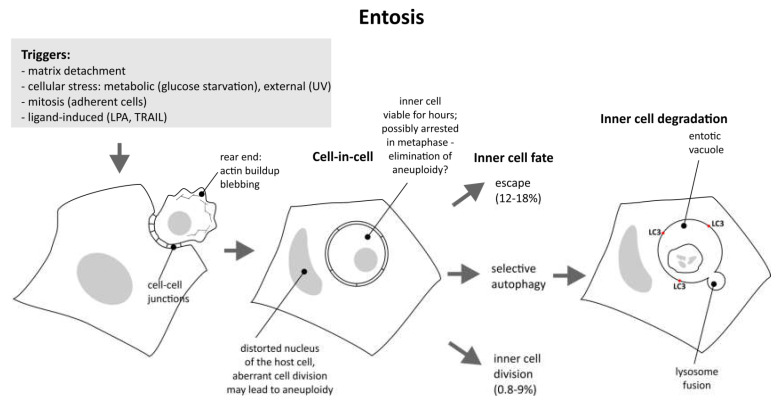
Entosis from initiation to completion; various triggers induce cytoskeletal changes, resulting in the formation of cell-in-cell structure. Both internal and external cells are prone to aneuploidy. Viable internal cells may still escape or divide, but the majority are destroyed by autophagy and lysosome fusion.

**Table 1 ijms-23-04985-t001:** Different types of cell-in-cell structures. Inner cell invasion can proceed homotypically, with epithelial cells of the same type (entosis), or heterotypically, when the invading cell is of a different type (usually leukocyte). The other type of CIC is formed when the outer cell actively engulfs the inner cell, in a process resembling endocytosis.

**Invasive CIC (Inner Cell Invasion)**
**Entosis:** A homotypic invasion of an epithelial cell into the cytoplasm of another epithelial cell. The host cell forms a vacuole around the inner cell, which deforms its own organelles, including nucleus. The inner cell can survive in the host cell for 12 h or longer and is eventually digested by the outer cell via an autophagosomal mechanism or escapes from the outer cell.
**Emperipolesis:** A heterotypic cell-in-cell structure, usually involving immune cells but also observed in tumor cells, in which a leukocyte is the inner cell (e.g., neutrophils in megakaryocytes or thymocytes in thymic nurse cells). **Emperitosis:** A type of emperipolesis in which the invading immune cell is cytotoxic (e.g., NK killer cells in tumor cells). It occurs with granzyme B-triggered apoptotic death of internalized killer cells. entry 2.
**Endocytic CIC (Engulfment by Outer Cell)**
**Phagocytosis-Like:** • Engulfment and digestion of a viable cell by macrophage **(Phagoptosis)** • Non-phagocytic cell acquires phagocytic phenotype
**Pinocytosis-Like:** Engulfment of CD4+ T lymphocytes (with a preference for regulatory T cells) by hepatocytes **(Enclysis)**

**Table 2 ijms-23-04985-t002:** Different types of cell death, triggered by different factors and occurring via different molecular mechanisms, with various levels of associated inflammation.

**Apoptosis:** Programed cell death occurring with chromatin condensation and DNA fragmentation has two activation mechanisms: intrinsic pathway (intracellular signals, which lead to changes in the inner mitochondrial membrane potential, release of pro-apoptotic factors into the cytosol and activation of caspase-9) and extrinsic pathway (receptor-induced activation of caspase-8), which induce apoptosis through caspase cascade. Proceeds without inflammation. **Anoikis:** A type of apoptosis occurring in response to the loss of attachment to extracellular matrix.
**Autophagic cell death:** Occurs in the absence of chromatin condensation but is accompanied by large-scale autophagic vacuolization of the cytoplasm and lysosomal degradation. A part of the general mechanism serving in the removal of damaged or unnecessary components, important for balancing sources of energy.
**CIC-type:** Cell death occurring with the formation of the cell-in-cell structure, with one viable cell fully inserted into the cytoplasm of another cell. See Table 1.
**Ferroptosis:** An iron-dependent cell death that occurs with iron accumulation and uncontrolled lipid peroxidation. Ferroptosis-inducing factors affect glutathione peroxidase, resulting in diminished antioxidant capacity, which leads to oxidative stress and cell death.
**Necrosis:** Premature cell death by autolysis resulting from an injury. Occurs with changes in the structure of the nucleus, formation of blebs and the rupture of the cell membrane, followed by an uncontrolled leaking of the cell’s content and a massive inflammatory response.
**Necroptosis:** Regulated necrosis mediated by death receptors, occurs with activation of receptor-interacting protein kinase 1/3 (RIPK1/3) and its substrate MLKL (mixed lineage kinase domain-like). Inflammatory.
**Parthanatos:** A type of cell death dependent on the activity of poly (ADP-ribose)-polymerase (PARP), triggered by nuclear translocation of the mitochondrial-associated apoptosis-inducing factor (AIF), occurs with chromatin condensation and DNA fragmentation. Does not require caspase activation.
**Phagocytosis:** A type of cell death in which the cell is phagocytosed (engulfed and degraded in a compartment called phagosome) by another cell, specialized (i.e., macrophage) or not specialized (i.e., tumor cell). Cell death by phagocytosis represents a specific application of the general mechanism designed to remove pathogens and cell debris.
**Pyroptosis:** An inflammatory programed cell death, which has similar features to apoptosis, such as caspase cascade, DNA damage and nuclear condensation. It occurs mostly after infection with intracellular pathogens, stroke, heart attack or cancer.

## Data Availability

Not applicable.

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
