# Peer review of "Cell Death by Entosis: Triggers, Molecular Mechanisms and Clinical Significance"

_ijms, 2022, doi:10.3390/ijms23094985_

Round 1

Reviewer 1 Report

This is an interesting review on the emerging field of entosis which provides a quite exhaustive view of all the contexts in which entosis was described and try to dissect putative physiological and pathological functions. It seems to be quite complete and cover the main current question of the field while providing interesting insights.  I have some minor suggestions that may help to facilitate reading and that may clarify some conceptual points.

  1. The role of cell stifness: While many part of the review turn-around this point, I think it would help to clearly state the essential role of differential stifness/contractility in the entosis process quite early in the review (including a brief description of these mechanical concepts). This will help to interpret all the results described in the first part of the manuscript that relate changes in GTPases, or mitosis (which is also well known to change coritcal stifness, see below), with engulfment. Along this line, there is a very interesting reference that describe a process akin to entosis in mice embryo and use 3D simulation to show that differential tension is enough to recapitulate this process (Maitre JL et al, Nat Cell Biol 2016 Hiiragi lab). I think it may help to add this reference.
  2. Along this line, given the role of mitosis in entosis, it would be important to refer to papers that clearly showed that cell stiffens during mitosis (through modulation of the cortex properties, eg : by activating ERM protein, see for instance different articles for the group of Buzz Baum, Kunda et al Curr Biol 2008 and many others). I think this may help to conceptually link mitosis, entosis and the important role of differential stifness
  3. Conceptually, entosis may be favour provided that the affinity between the engulfed cell and its neigbours is somewhat higher than the affinitiy with the other neighbours cells and the affinity for the substrate. I think it may help to intereprest some of the results along these lines (which more or less appear in between the lines, but it may need to be stated clearly somewhere).
  4. It is not always clear, especially in the first part of the manuscript, whether the conditions described are affecting the entry of the cell (so the early phase of entosis) versus later phases which determine the fate of the inner cell (death or exit). I am wondering whether it would not help to clearly delineate these steps from the beginning (as performed in the later part of the manuscript number 5) before listing all the conditions affecting/triggering entosis ? 
  5. To facilitate the reading, it may be good to add a few sentences at the end of each part that sum up the main concepts that can be outlined. This may help to connect the different part and help to synthetise the main message.

Other minor points : 

Page 3: E-cad section, “This is consistent with the finding that IL-8 depletion caused….” : for clarity you may want to explain whare are CDH3 and CTNNG (explain that this is a P-cadherin and junctional plaque components)

Line 194: typo, inner call

Author Response

This is an interesting review on the emerging field of entosis which provides a quite exhaustive view of all the contexts in which entosis was described and try to dissect putative physiological and pathological functions. It seems to be quite complete and cover the main current question of the field while providing interesting insights.  I have some minor suggestions that may help to facilitate reading and that may clarify some conceptual points.

    The role of cell stifness: While many part of the review turn-around this point, I think it would help to clearly state the essential role of differential stifness/contractility in the entosis process quite early in the review (including a brief description of these mechanical concepts). This will help to interpret all the results described in the first part of the manuscript that relate changes in GTPases, or mitosis (which is also well known to change coritcal stifness, see below), with engulfment. Along this line, there is a very interesting reference that describe a process akin to entosis in mice embryo and use 3D simulation to show that differential tension is enough to recapitulate this process (Maitre JL et al, Nat Cell Biol 2016 Hiiragi lab). I think it may help to add this reference.

The paragraph(s) concerning the role of stiffness/contractility and the reference were added.

    Along this line, given the role of mitosis in entosis, it would be important to refer to papers that clearly showed that cell stiffens during mitosis (through modulation of the cortex properties, eg : by activating ERM protein, see for instance different articles for the group of Buzz Baum, Kunda et al Curr Biol 2008 and many others). I think this may help to conceptually link mitosis, entosis and the important role of differential stiffness

Cell  stiffness during mitosis was addressed and the appropriate reference was added.

    Conceptually, entosis may be favour provided that the affinity between the engulfed cell and its neigbours is somewhat higher than the affinitiy with the other neighbours cells and the affinity for the substrate. I think it may help to intereprest some of the results along these lines (which more or less appear in between the lines, but it may need to be stated clearly somewhere).

    It is not always clear, especially in the first part of the manuscript, whether the conditions described are affecting the entry of the cell (so the early phase of entosis) versus later phases which determine the fate of the inner cell (death or exit). I am wondering whether it would not help to clearly delineate these steps from the beginning (as performed in the later part of the manuscript number 5) before listing all the conditions affecting/triggering entosis ?

The corrections were introduced, including delineation of phases.

    To facilitate the reading, it may be good to add a few sentences at the end of each part that sum up the main concepts that can be outlined. This may help to connect the different part and help to synthetise the main message.

Other minor points :

Page 3: E-cad section, “This is consistent with the finding that IL-8 depletion caused….” : for clarity you may want to explain whare are CDH3 and CTNNG (explain that this is a P-cadherin and junctional plaque components)

Line 194: typo, inner call

The corrections were made.

Reviewer 2 Report

In the current manuscript entitled "Cell death by entosis; triggers, molecular mechanisms, and clinical significance", In this review, the authors summarize multiple triggers, molecular signaling, and clinical significance. However, there was not notable comments in the “Discussion points” section. Also, please comment on entosis and chemotherapy.

Martins I, et al. Anticancer chemotherapy and radiotherapy trigger both non-cell-autonomous and cell-autonomous death. Cell Death Dis. 2018;9(7):716.

Tonnessen-Murray CA, et al. Chemotherapy-induced senescent cancer cells engulf other cells to enhance their survival. J Cell Biol. 2019;218(11):3827-3844.

Hayashi A, et al. Genetic and clinical correlates of entosis in pancreatic ductal adenocarcinoma. Mod Pathol. 2020;33(9):1822-1831.

Liu J, et al. Induction of entosis in prostate cancer cells by nintedanib and its therapeutic implications. Oncol Lett. 2019;17(3):3151-3162.

Author Response

In the current manuscript entitled "Cell death by entosis; triggers, molecular mechanisms, and clinical significance", In this review, the authors summarize multiple triggers, molecular signaling, and clinical significance. However, there was not notable comments in the “Discussion points” section. Also, please comment on entosis and chemotherapy.

Martins I, et al. Anticancer chemotherapy and radiotherapy trigger both non-cell-autonomous and cell-autonomous death. Cell Death Dis. 2018;9(7):716.

Tonnessen-Murray CA, et al. Chemotherapy-induced senescent cancer cells engulf other cells to enhance their survival. J Cell Biol. 2019;218(11):3827-3844.

Hayashi A, et al. Genetic and clinical correlates of entosis in pancreatic ductal adenocarcinoma. Mod Pathol. 2020;33(9):1822-1831.

Liu J, et al. Induction of entosis in prostate cancer cells by nintedanib and its therapeutic implications. Oncol Lett. 2019;17(3):3151-3162.

The text was modified and the references were introduced (except Hayashi et al. which was already cited).